# A novel cytoskeletal action of xylosides

**Caitlin P. Mencio**[1¤], **Sharada M. Tilve**[1], **Masato Suzuki**[2], **Kohei Higashi**[2], **Yasuhiro Katagiri**[1], **Herbert M. Geller**[1]*

**1** Laboratory of Developmental Neurobiology, National Heart, Lung, and Blood Institute, National Institutes of Health, Bethesda, MD, United States of America, **2** Faculty of Pharmaceutical Sciences, Tokyo University of Science, Noda City, Chiba, Japan

¤ Current address: Schubot Center for Avian Health, Department of Veterinary Pathobiology, College of Veterinary Medicine & Biomedical Sciences, Texas A&M University, College Station, TX, United States of America

* gellerh@nhlbi.nih.gov

**Data Availability Statement:** All relevant data for Figs 1–5 are within the paper and its Supporting Information files. Data supporting Fig 6 are available at https://www.ncbi.nlm.nih.gov/geo/query/acc.cgi?acc=GSE206057.

## Abstract

Proteoglycan glycosaminoglycan (GAG) chains are attached to a serine residue in the protein through a linkage series of sugars, the first of which is xylose. Xylosides are chemicals which compete with the xylose at the enzyme xylosyl transferase to prevent the attachment of GAG chains to proteins. These compounds have been employed at concentrations in the millimolar range as tools to study the role of GAG chains in proteoglycan function. In the course of our studies with xylosides, we conducted a dose-response curve for xyloside actions on neural cells. To our surprise, we found that concentrations of xylosides in the nanomolar to micromolar range had major effects on cell morphology of hippocampal neurons as well as of Neuro2a cells, affecting both actin and tubulin cytoskeletal dynamics. Such effects/morphological changes were not observed with higher xyloside concentrations. We found a dose-dependent alteration of GAG secretion by Neuro2a cells; however, concentrations of xylosides which were effective in altering neuronal morphology did not cause a large change in the rate of GAG chain secretion. In contrast, both low and high concentrations of xylosides altered HS and CS composition. RNAseq of treated cells demonstrated alterations in gene expression only after treatment with millimolar concentration of xylosides that had no effect on cell morphology. These observations support a novel action of xylosides on neuronal cells.

## Introduction

Proteoglycans (PGs) are found in every tissue in the body. Consisting of two major components, a core protein and glycosaminoglycan (GAG) chain(s), they are essential components of the extracellular matrix. There are three major classes, differentiated by their GAG chain composition: heparan sulfate (HS), chondroitin sulfate (CS) and keratan sulfate (KS). PGs are involved in many important biological processes [1], especially in the central nervous system, where they regulate neuronal migration, axon guidance and differentiation [2, 3]. They are also believed to play critical roles in neural de- and re-generation [4].

In the recent past, it has become more apparent that the physiological effects of proteoglycans can be attributed to their GAG chains. Moreover, the wide range of biological function is

**Funding:** This work was funded by the NHLBI Division of Intramural Research (DIR) with project number 1Z01HL006021. The funders had no role in study design, data collection and analysis, decision to publish, or preparation of the manuscript.

**Competing interests:** The authors have declared that no competing interests exist.

often attributed to the complexity and diversity of GAG chain modifications, sulfation being the most common. Altering GAG sulfation patterns has been known to change GAG chain receptor binding resulting in modified cell signaling [5–7].

GAG chain biosynthesis is a non-template driven process that begins for all PGs with a common linkage of three sugars (Xyl-Gal-Gal) to a serine on the core protein. The three classes then diverge with addition of disaccharides to the linkage region. For HS, these would be GlcNAc and GlcA, for CS the disaccharides are GalNAc and GlcA, and for KS they are GlcNAc and Gal. Each disaccharide in the chain may undergo several different modifications, primarily sulfation. The impact that modifications to the GAG chains has on neural development as well as other cellular processes remains an active area of research.

One major approach to understanding the function of GAG chains in PGs has been by using chemical modulators of GAG biosynthesis called xylosides which serve as competitive molecules for cellular sugar chain synthesis. Xyloside-induced interference in GAG biosynthesis can lead to changes in GAG concentration, chain type, sulfation pattern, and molecular weight [8–10] as well as biological activity [11–13]. Historically, xyloside research has utilized high (millimolar) concentrations that serve to primarily block endogenous GAG production [14, 15], with relatively few studies using lower than mM concentrations [10, 16]. We therefore conducted a dose-response study to determine the minimal concentration of xyloside that would disrupt GAG chain synthesis in hippocampal pyramidal neurons and Neuro2a cells. To our surprise, we saw major changes in cell morphology when cells were treated with nanomolar xyloside concentrations but not micro- or millimolar treatment. After documenting the cytoskeletal changes, we then determined that treatment of cells with nanomolar concentrations of xyloside altered CS and HS composition with distinct differences found in heparan sulfate (HS) disaccharide composition from both control cells and cells treated with the high concentration of xyloside. This study serves as a first step in our attempt to understand how minor shifts in GAG chain composition or concentration can affect biological processes and continues to support the critical role of sugars in development and cellular function.

## Materials and methods

### Laboratory animals

Experiments and procedures were performed in accordance with Institutional Animal Care and Use Committee (IACUC) at the National Institutes of Health approved protocols. Pregnant female C57Bl/6 mice (Charles River) were housed in a pathogen free facility with standard 12 h light/dark cycle and unlimited access to food and embryonic (E17-19) pups were utilized for embryonic hippocampal neuron primary cultures.

### Cell culture

Primary hippocampal neuron cultures were prepared from embryonic (E17-19) C57Bl/6 mouse brains. Hippocampi were dissected and dissociated into single cell suspensions. Dissociated cells were seeded onto coverslips coated with poly-L-lysine and cultured in 500 μL Neurobasal medium containing B27 supplement (Thermo Fisher) and 24 mM KCl at a density of $8–10 \times 10^3$ cells/well to allow for observation of isolated neurons. After allowing 2 h for neuronal attachment, media was replaced with 1 ml fresh Neurobasal media containing DMSO and concentrations of 4-Methylumbelliferyl-β-D-xylopyranoside (4-MU, SigmaAldrich) between 0.5 nM and 1 mM. Xylosides were dissolved in DMSO and stock solutions of 1 M and 500 μM were made. DMSO and xylosides were added to cells at appropriate dilutions resulting in final concentrations of xyloside and 0.1% DMSO in solution. Cells were incubated for 24–72 h at

37˚C and 5% CO2 atmosphere and then fixed and stained for DAPI, βIII-tubulin, and actin (Phalloidin).

Neuro2a (ATCC) cells were cultured in DMEM media containing 10% fetal bovine serum (FBS) at 37˚C with 5% $CO_2$. Cells were seeded onto coverslips, 35 mm glass bottom dishes (MatTek), 6-well plates or T-75 (Corning) flasks depending on experimental design. Cells were seeded at ~8–10 ×$10^3$ cells/well for coverslips, 35mm dishes and 6-well plates and ~1.5 ×$10^6$ cells/flask for T-75 flasks.

To produce conditioned media, Neuro2a cells were grown to about 50% confluency in T-75 flasks (Corning) in DMEM supplemented with 10% FBS. At this point, DMEM containing FBS was removed and cells were washed twice with sterile PBS and then kept in DMEM only overnight. The next day the media was replaced with media containing DMSO, 500 nM xyloside or 1 mM xyloside. Cells were allowed to grow for 48 h, after which time media was removed and placed into 15 ml conical tubes and spun for 10 min at 500 rpm to remove floating cells and debris. The supernatant was transferred into a fresh tube and subject to GAG analysis.

## Glycosaminoglycan analysis

Conditioned media was harvested from Neuro2a cells that had been serum starved overnight and then treated for 48 h with xylosides in DMEM culture medium. Approximately 10 ml of conditioned media was collected from each experimental condition: DMSO, 500 nM and 1 mM xyloside. This media was spun for 5 min at 2000 rpm to remove cell debris and then transferred to a new conical tube and frozen until analysis.

GAG extraction was performed as follows. The media (1 mL) was treated with 10% TCA and centrifuged at 12000 rpm for 5 min to remove proteins. GAGs were collected by Amicon Ultra Centrifugal Filter 3K device (Merck Millipore, Billerica, MA, USA) and suspended with 100 μL of $H_2O$. Fifty μl of GAG solution was moved to new 1.5 ml microcentrifuge tube and lyophilized. Resulting GAG samples were incubated in the reaction mixture (35 μL) containing 28.6 mM Tris-acetate (pH 8.0) and 50 mIU of chondroitinase ABC for 16 h at 37˚C. Depolymerized samples were boiled and evaporated, unsaturated disaccharides of CS were collected by Amicon Ultra Centrifugal Filter 30K device (Merck Millipore). The remaining HS samples in filters of spin columns were transferred to new microtubes and incubated in 16 μl of reaction mixture (pH 7.0), containing 1 mU heparinase I (Seikagaku Corp., Tokyo, Japan), 1 mU heparinase II (Iduron, Manchester, UK), 1 mU heparinase III (Seikagaku), 31.3 mM sodium acetate, and 3.13 mM calcium acetate for 16 h at 37˚C.

Unsaturated disaccharide analysis using reversed phase ion-pair chromatography with sensitive and specific post-column detection was performed as described previously [17]. Disaccharide composition analysis of CS or HS was performed by reversed phase ion-pairing chromatography with sensitive and specific post-column detection. A gradient was applied at a flow rate of 1.0 ml min$^{-1}$ on Senshu Pak Docosil (4.6 × 150 mm; Senshu Scientific Co., Ltd., Tokyo, Japan) at 60˚C. The eluent buffers were as follows: A, 10 mM tetra-*n*-butylammonium hydrogen sulfate in 12% methanol; B, 0.2 M NaCl in buffer A. The gradient program of CS disaccharides analysis was as follows: 0–10 min (1% B), 10–11 min (1–10% B), 11–30 min (10% B), 30–35 min (10–60% B), and 35–40 min (60% B). The gradient program of HS disaccharides analysis was as follows: 0–10 min (1–4% B), 10–11 min (4–15% B), 11–20 min (15–25% B), 20–22 min (25–53% B), and 22–29 min (53% B). Aqueous (0.5% (w/v)) 2-cyanoacetamide solution and 1 M NaOH were added to the eluent at the same flow rates (0.25 ml min$^{-1}$) by using a double plunger pump. The effluent was monitored fluorometrically (Ex., 346 nm; Em., 410 nm). Expression levels of HS or CS were expressed as total amounts of unsaturated disaccharides, while the composition was expressed as percent of total GAG.

## Microscopy and image processing

Cells were imaged using either a Nikon A1R or Zeiss 880 confocal microscope with 60X and 63X objectives depending on the experiment. Z-stacks were maximally projected onto a single plane using Zeiss or ImageJ [18] image processing software. For images used in fluorescence quantification, image capture settings were held constant, and samples from within each group were imaged at the same time. Fluorescence intensity was measured using ImageJ with identical settings for all samples within each analysis.

## Neurite outgrowth and growth cone analysis

After fixation and staining, at least 60 images were taken across two coverslips per condition. Files were analyzed by an experimenter blinded to the experimental conditions. Neurons were measured if they were isolated from other neurons and had distinct nuclei and at least one neurite longer than the diameter of the cell body. At least 60 images were taken for each experimental condition and then randomized using the "Bulk Rename" utility (https://www.bulkrenameutility.co.uk). Duplicates of these files, with all identifying information removed were analyzed. After analysis, results were matched to the treatment condition. Neurite looping at growth cones was counted in a binary yes/no fashion based on the presence of visible circular strands of microtubules (MTs) at the terminal end of the longest neurite. These loops required the presence of an empty center surrounded by one or more continuous MT. If a loop was observed the cell was counted as "having looped MTs", otherwise it was counted as "not". Neurite measurements were obtained using the ImageJ trace tool measuring the distance from where the neurite connected to the soma and terminating at the end of the neurite. Both longest and total neurite measurements were obtained for each neuron.

Growth cone measurements were conducted using ImageJ by tracing the end of the neurite from the point it widened from its average diameter of the stalk and where phalloidin staining began to show more intense and spiked appearance as is common with growing ends of neurites. Only the largest growth cone of the neuron was measured. Each experiment was performed in triplicate.

## Analysis of cytoskeleton dynamics

EB3 comet analysis: Neuro2a cells were grown in culture until about 30% confluent. Cells were then treated with DMSO, 500 nM xyloside or 1 mM xyloside for 48 h. After a further 24 h, cells were transfected with EB3-GFP using Avalanche®-Omni transfection reagent [19]. The next day, cells were imaged on a Nikon A1R confocal microscope with a 60X/1.42 N.A. Plan Apochromat oil objective. Transfected cells were imaged every 12.5 sec over 5 min. Images were then processed using u-track software [20] and comets assessed for speed, lifetime and distance travelled.

Actin bundle analysis: Neuro2a cells were grown in culture until about 30% confluent. Cells were then treated with DMSO, 500 nM xyloside or 1 mM xyloside for 48 h. After a further 24 h, cells were transfected with Ftractin-mCherry using Avalanche®-Omni transfection reagent. The next day, cells were imaged using a Zeiss 880 confocal microscope with a 63X/1.4NA apochromat objective. Transfected cells with lamellipodia were imaged in a z-stack. Using ImageJ, z-stacks were max projected, and a line drawn through the lamellipodia and a line scan performed based upon fluorescence. Actin bundles were identified as peaks of increased fluorescence. Bundles were counted and the area under the curve taken to compare quantity and size of the bundles.

### RNA-seq

Between 6–15 micrograms per sample of total RNA from three samples each of Neuro2a cells treated with either DMSO, LCX or HCX were sequenced by Illumina at Omega Bioservices (Norcross, GA). Data analysis was performed using the Partek Flow statistical analysis software (Partek Incorporated). For this, raw data from three replicates of the three conditions were imported to Partek Flow for alignment and quality controls. Aligned reads were quantified to transcriptome, filtered out on low expression and normalized. Feature gene lists with at least a two-fold change in gene expression with a false discovery rate of Q < 0.05 using the BH procedure [21] were created by pairwise comparison. Pseudogenes were eliminated from the final list of altered genes and the list plotted as a heatmap. In order to visualize the difference of the expression between DMSO, LCX and HCX, the data were centered by subtracting the mean of the $\log_2$ Fold Change of all samples for each gene from the original $\log_2$ Fold Change value. The centered data of all samples were then plotted into a heat map. The rows of the heatmaps (genes) were ordered by fold change [22].

### Statistics

All statistical tests were performed using GraphPad Prism 7.0 (GraphPad Software, La Jolla, CA). Neurite lengths in culture were compared using Kruskal-Wallis Analysis of Variance and Mann-Whitney U tests.

## Results

Our objective was to determine the effect of xyloside treatment on the morphology of hippocampal neurons in culture. A previous study noted that concentrations of 0.1 and 0.2 mM xyloside perturbed the generation of neuronal polarity [23], but we sought to determine a more complete dose-response curve. We therefore added 4-MU in a range of 0.5 nM to 1 mM to the medium of dissociated embryonic hippocampal neurons, and fixed and stained cultures after 24 or 72 h. At 72 h, we observed a surprising dose-dependent response to 4-MU: neurons treated with lower concentrations showed enlarged growth cones that exhibited looped microtubules as compared to DMSO treated cultures or cultures treated with mM concentrations (Fig 1, S1 Fig). Fig 1A shows representative images taken from the range of concentrations: DMSO (control), 500 nM 4-MU as the low xyloside concentration (LCX) and 1 mM MU as the high xyloside concentration (HCX). As presented in Fig 1B, there was a difference in growth cone size with the different treatments: neurons treated with LCX had significantly larger growth cones than those treated with DMSO or HCX. To determine if the enlarged growth cones was due to neurite stalling, the length of the longest neurite was measured for each neuron in the condition. There was no significant difference in neurite length between all conditions (Fig 1C). The lower end of the dose-response curve is presented in S1 Fig. Moreover, a difference in neurite morphology was apparent as early as 24 hr. after plating (S2 Fig), where neurites of LCX-treated neurons showed splayed microtubules, while neurites of cells treated with either DMSO or HCX show typical compact neurite outgrowth.

As primary neuron culture is often limiting in cell number, we decided to assess if the mouse neural crest-derived cell line Neuro2a would show similar morphological changes and could be used to study any structural or biochemical changes between HCX and LCX treatment. Observation of the cytoskeleton and overall cellular morphology in LCX treated Neuro2a cells shows that these cells exhibit actin-rich lamellipodia that were not observed in Neuro2a cells treated with either DMSO or HCX (S3 Fig). With a confirmed change in cytoskeleton and for the sake of uniformity and economy, we decided to utilize Neuro2a cells for all other experiments in this study.

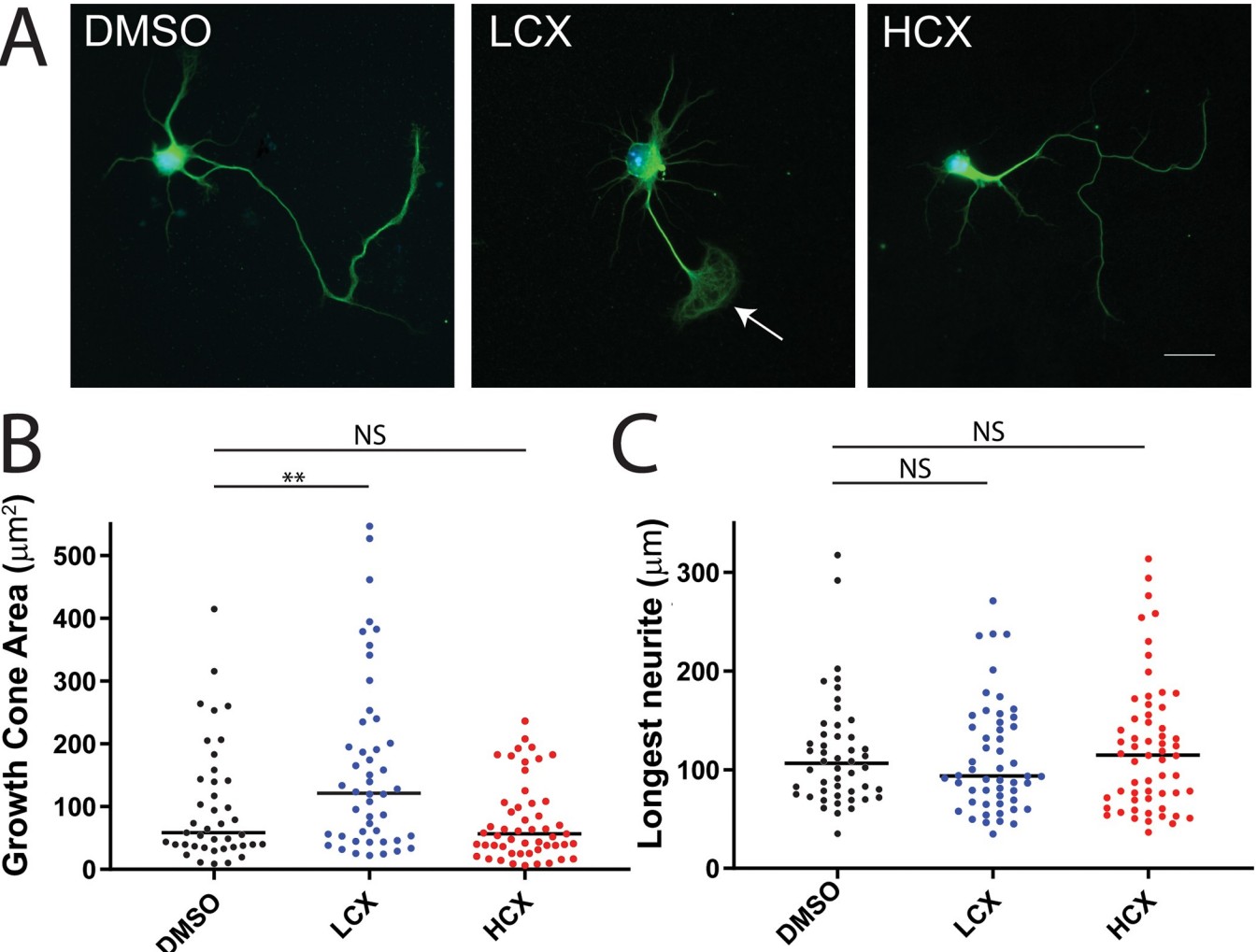

**Fig 1. Altered cytoskeleton seen in cells treated with 500 nM (LCX) but not 1 mM (HCX) xyloside.** A) Example of a primary mouse hippocampal neuron treated for 72h with DMSO, LCX, or HCX and stained with anti-βIII-tubulin. White arrow indicates enlarged growth cone in LCX treated neuron. B) Quantification of growth cone area and length of longest neurite.in DMSO, LCX and HCX treated neurons. LCX treated neurons have significantly increased growth cone area compared to control or HCX conditions. Scale = 25 μm. $^{**}p < 0.01$.

## LCX treatment reduces cellular movement

After observing changes in morphology, we next wanted to check if these changes alter the cells' ability to migrate. Neuro2a cells were treated with DMSO, LCX or HCX for 48 h, at which point cells were imaged using brightfield microscopy for 4 h with images being taken every 8–10 min (Fig 2). Cells within the image were tracked and velocity and total distance was calculated. There was a significant reduction in both velocity and total distance in LCX treated cells as compared to DMSO and HCX treated Neuro2a cells (Fig 2B). This reduction in movement implicates possible changes in cytoskeleton dynamics.

## LCX treatment affects both actin and microtubule dynamics

To determine if altered cytoskeleton may play a role in LCX induced changes in morphology and migration, we next assessed both actin and microtubules in Neuro2a cells. Neuro2a cells were transfected with Ftractin-mCherry and treated for 48h with DMSO, LCX or HCX. At

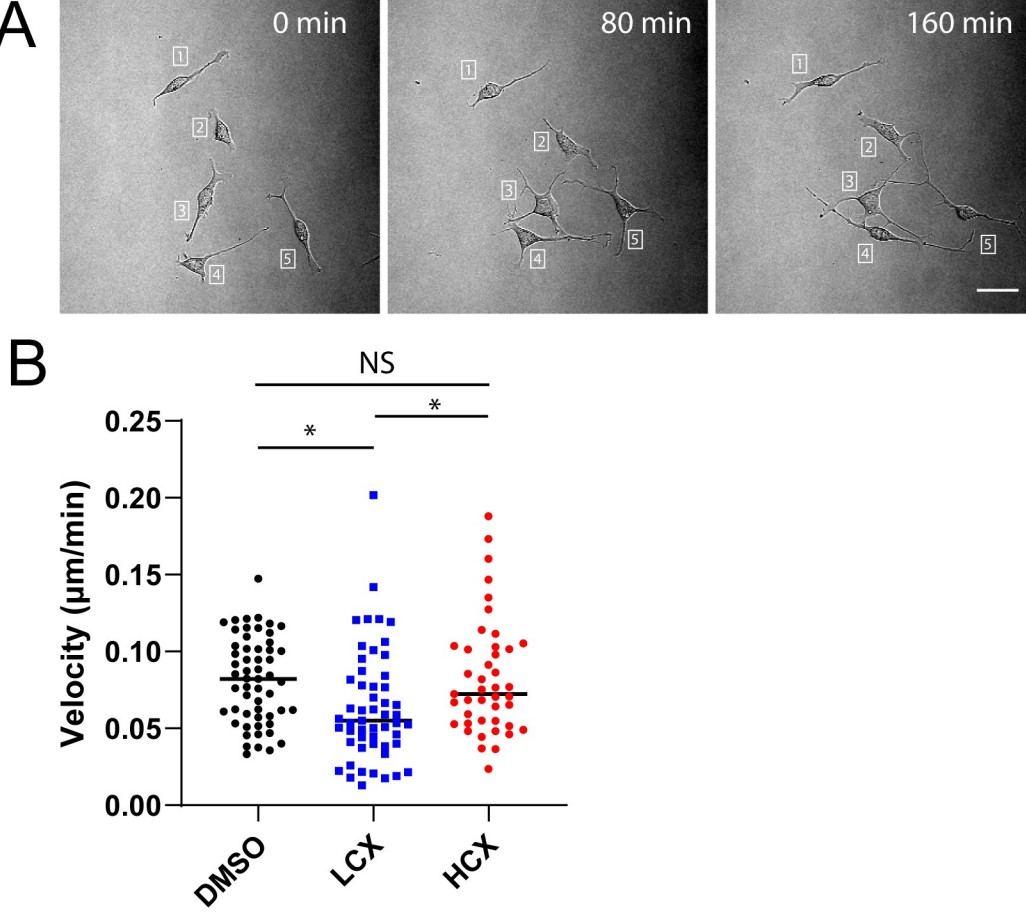

**Fig 2. LCX reduces cell migration.** Neuro2a cells were observed using time-lapse microscopy. A) Representative images taken at the indicated intervals. Numbers show positions of specific cells. B) Dot plot of velocity of individual cells taken as distance moved in each frame. *p < 0.05. **p < 0.01. Scale = 25 μm.

48h, visual observation showed a marked difference in lamellipodia of LCX treated cells as compared to DMSO and HCX treated Neuro2a cells. DMSO and HCX-treated cells showed bright and thick actin bundles while LCX treated cells appeared to have fewer bundles which also appeared thinner than their control counterparts (Fig 3A). We quantified both the area and number of actin bundles from these images. LCX treated Neuro2a cells exhibited lamellipodia that had significantly fewer actin bundles per 10μm when compared to controls (Fig 3B). Additionally, the area under the curve for these bundles was significantly reduced in LCX treated cells as compared to DMSO or HCX treated Neuro2a cells (Fig 3C) indicating less robust actin bundles in LCX lamellipodia.

With a measured effect on actin, we next wanted to assess if microtubules, the other major cytoskeleton element, were also affected. To examine microtubule dynamics, Neuro2a cells were treated for 48h with DMSO, LCX or HCX, and transfected with EB3-GFP as outlined in materials and methods. EB3-GFP cells were imaged at 60X with one image taken every 12.5 sec for 5 min (Fig 4A). Images were used to measure speed, persistence and total distance traveled of individual EB3 speckles. Both LCX and HCX treated Neuro2a cells exhibited a higher percentage of faster moving EB3 comets as compared to controls (Fig 4B, left). Additionally, xyloside treatment resulted in longer persistence of EB3 comets compared to DMSO treated

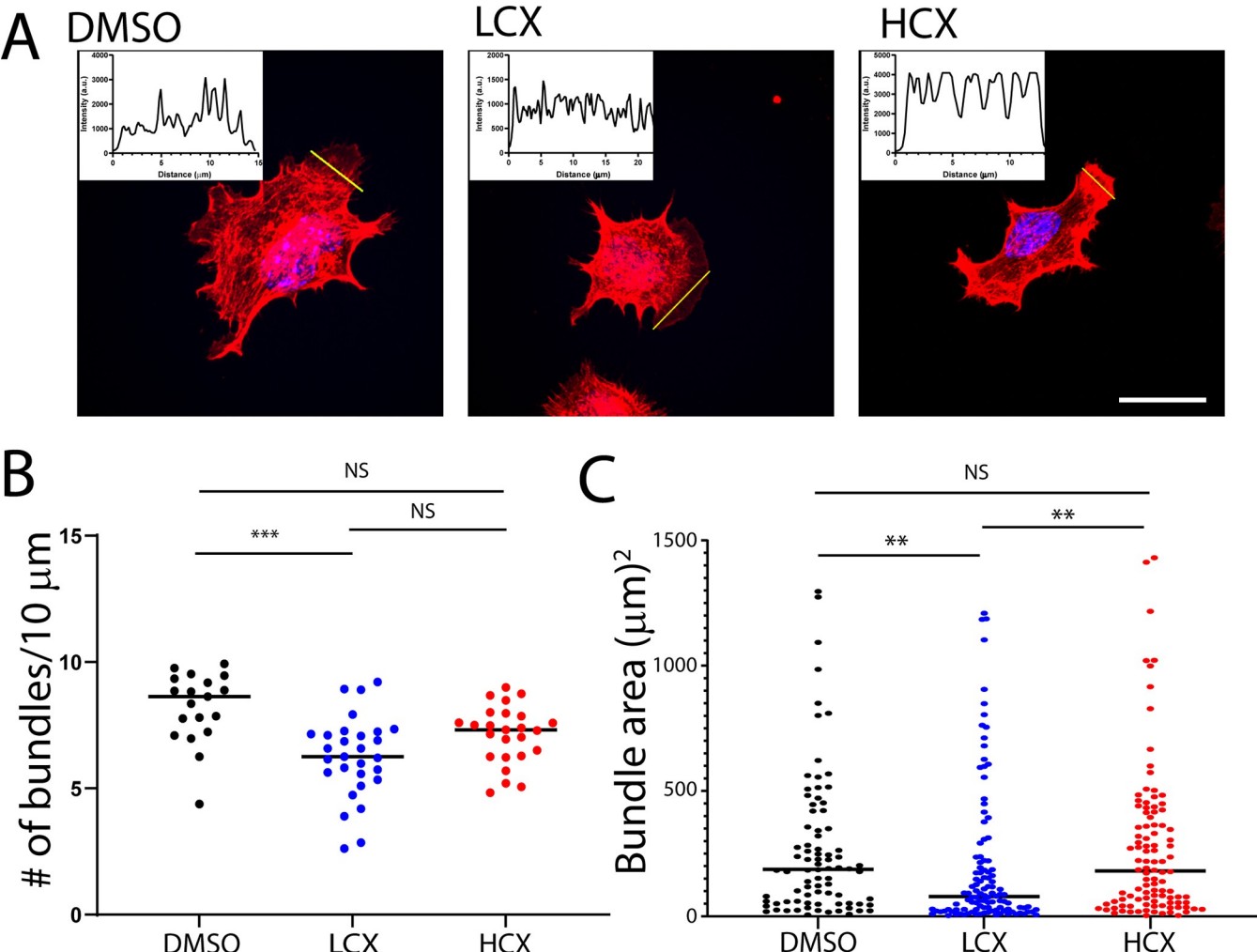

**Fig 3. LCX alters actin bundling.** A) Representative fluorescence images of living Neuro2a cells expressing F-tractin treated with DMSO, LCX, or HCX for 48 h. Insets show line scans of fluorescence intensity across the lamellipodia of indicated cells. B) Plots of area under the curve and the number of peaks for each of the three conditions. Scale = 25 μm. $^{**}$p < 0.01.

Neuro2a cells (Fig 4B, center). There were no significant changes in total distance traveled between all conditions (Fig 4B, right). Thus, while LCX has selective actions on actin dynamics, both LCX and HCX affected microtubule polymerization.

## LCX treatment alters GAG chain but not mRNA profile

Xylosides have been known to affect cellular GAG chains, both increasing the rate of synthesis and secretion [24] and altering GAG chain composition [25]. These GAG chains have been linked to cellular signaling that could lead to altered cytoskeleton. To determine if LCX treatment affects GAG chain production or sulfation, Neuro2a cells were treated with DMSO, LCX, or HCX in serum free media for 48h. Conditioned media were collected and the concentration as well as the composition of GAG chains were analyzed. Composition of the GAG addresses the positions of sulfate groups on the individual CS or HS disaccharides. As previously reported, there was a dose dependent increase in GAG chain accumulation for both CS and HS in the medium with xyloside treatment. Both LCX and HCX increased CS chain

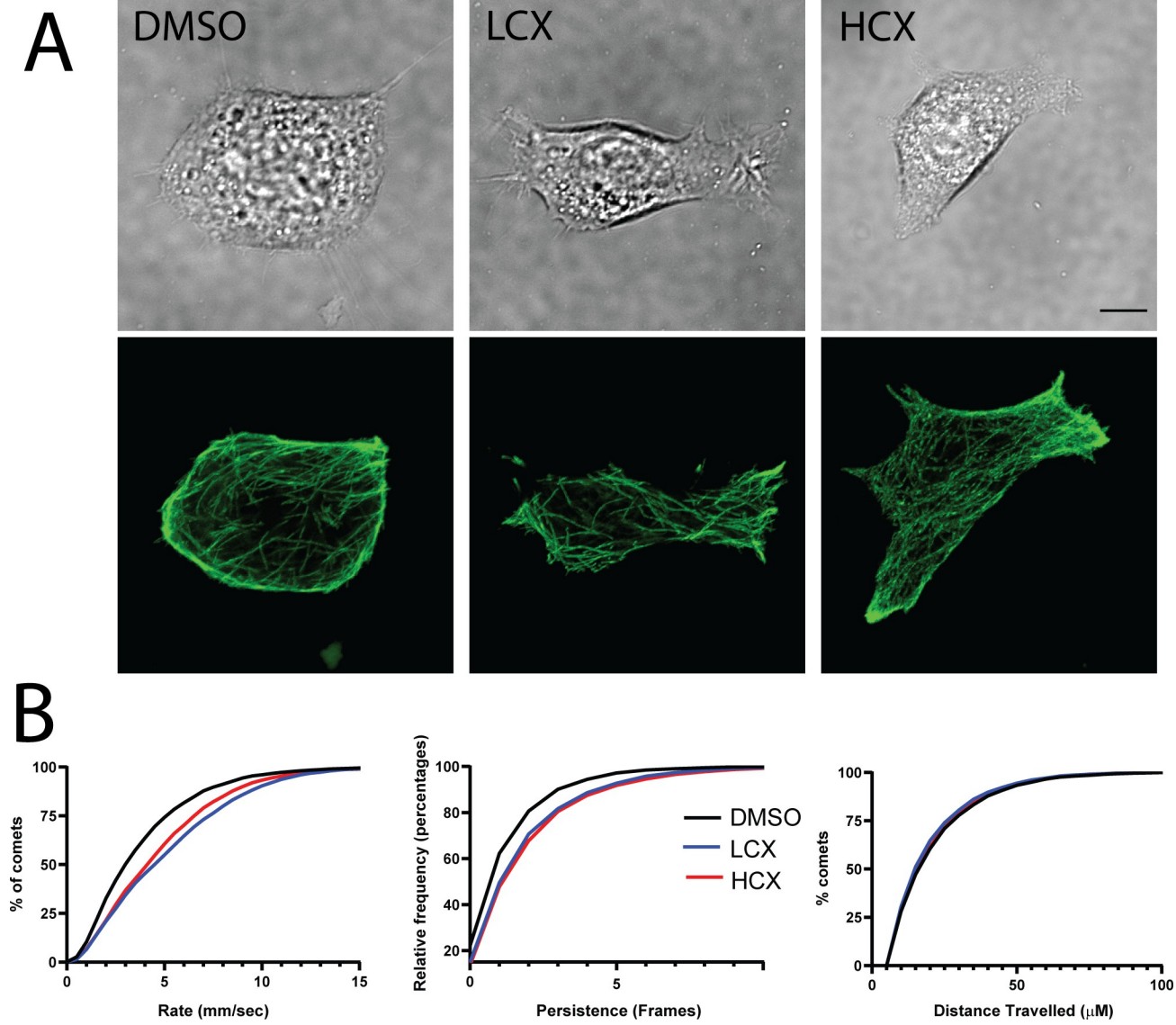

**Fig 4. Xyloside treatment alters microtubule dynamics.** A) Images of living Neuro2a cells (top) expressing EB3-EGFP were imaged over time. Bottom images show z-projections of 25 frames of fluorescent images of EB3-EGFP. B) Cumulative distribution plots of speed, persistence and total distance traveled by EB3 comets from DMSO, LCX and HCX-treated cells. Scale = 10 μm.

concentration as compared to control. Notably, HCX caused a much larger increase in CS (Fig 5A). In contrast, there was no difference between HS concentration in CM of DMSO and LCX-treated cells; only HCX treatment led to a large increase in HS secretion as compared to DMSO and LCX (Fig 5B).

We assayed the disaccharide composition of the secreted GAGs, both CS and HS. Conditioned media from DMSO treated cells contained a close to even split between non-sulfated and 4-sulfated CS GAG chains. Following xyloside treatment with both LCX and HCX, we observed an increase in 4-sulfated CS GAG (Fig 5C). When we examined the disaccharide composition of secreted HS, it was found that all three treatment groups had different disaccharide profiles. Conditioned media from DMSO treated cells showed mostly non-sulfated disaccharides with a small percentage of N-sulfated and even smaller group of 6-sulfated

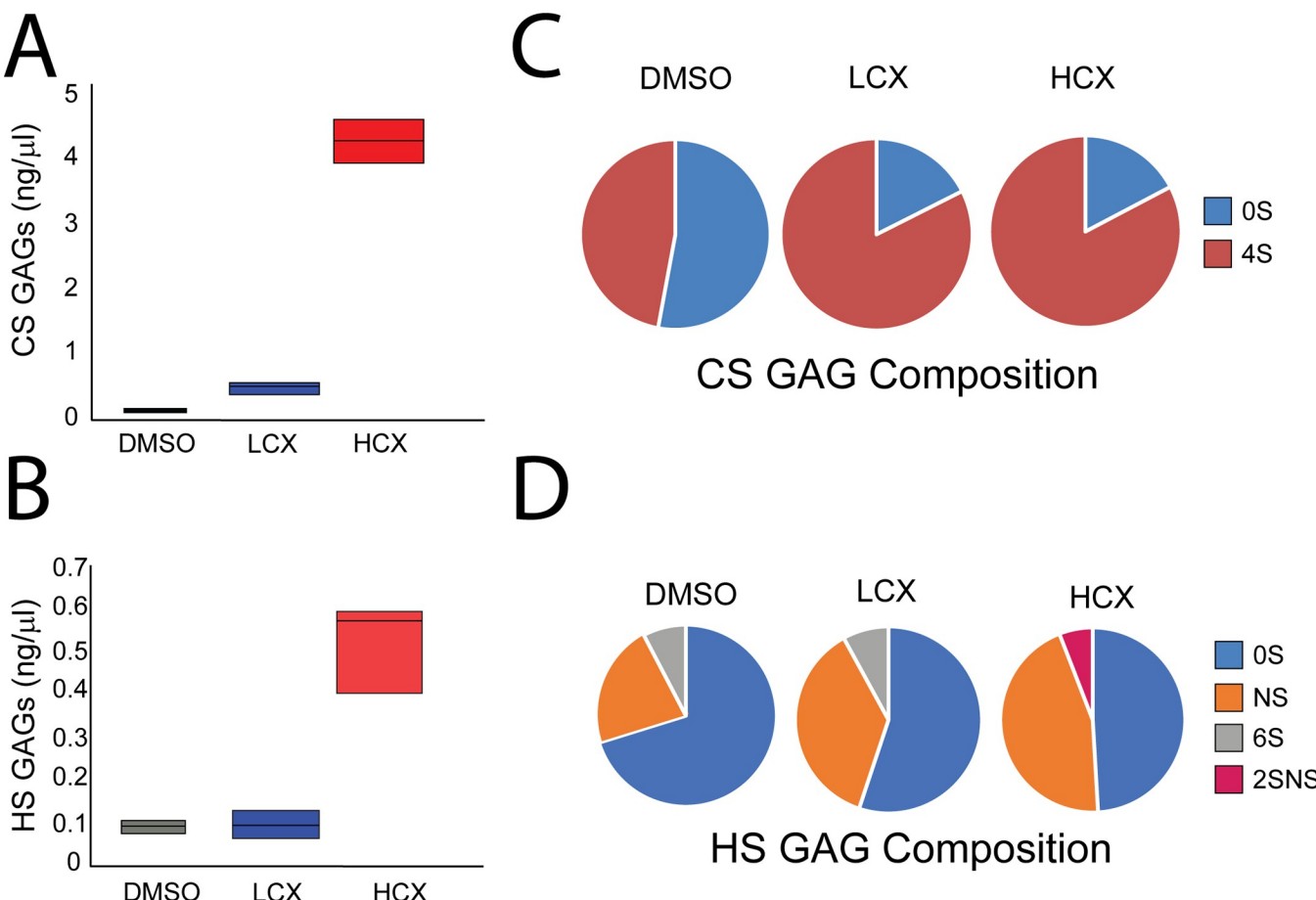

**Fig 5. Analysis of glycosaminoglycan chains in conditioned media from Neuro2a cells after xyloside treatment.** GAG chains were collected from conditioned medium and subject to analysis of total GAG levels as well as disaccharide composition, expressed as percentage of total GAGs. A) Quantification of CS GAG levels present in conditioned media sample. B) CS disaccharides analysis. 0S, ΔHexUA-GalNAc; 4S, ΔHexUA-GalNAc(4-O-sulfate). C) Quantification of HS GAG levels present in conditioned media sample. D) HS disaccharide analysis. 0S, ΔHexUA-GlcNH₂; NS, ΔHexUA-GlcNS; 6S, ΔHexUA-GlcNH₂(6-O-sulfate); 2SNS, ΔHexUA(2-O-sulfate)-GlcNS.

disaccharides (Fig 5D). In contrast, LCX treatment resulted in an increase in N-sulfated disaccharides while HCX treatment led to a larger increase in N-sulfation, loss of 6-sulfation and the presence of a small amount of 2-sulfated, N-sulfated disaccharides (Fig 5D). These findings indicate that xyloside treatment can change GAG chain synthesis and composition, and alterations to HS appear to occur in a concentration dependent manner.

Because xyloside effects on the cytoskeleton take several days to develop, it is possible that xylosides altered transcription which may have led to altered protein expression. Therefore, we next performed RNA-seq to determine which, if any, RNAs were altered by xyloside treatment. Cultures of Neuro2a cells were treated with DMSO, LCX and HCX and RNA was extracted after 48 h and pyrosequenced. The raw data were filtered using Partek Flow as noted in materials and methods. The filtered counts were then used to determine those genes whose expression was altered by >2X and statistically significant using q<0.05 FDR. Using these criteria, we found no genes whose change in expression was different between LCX and DMSO, while we did find changes in gene expression when comparing HCX with both DMSO and LCX. Fig 6A presents volcano plots which indicate the differentially expressed genes between HCX and DMSO and HCX and LCX. Fig 6B presents a Venn diagram that indicates the number of

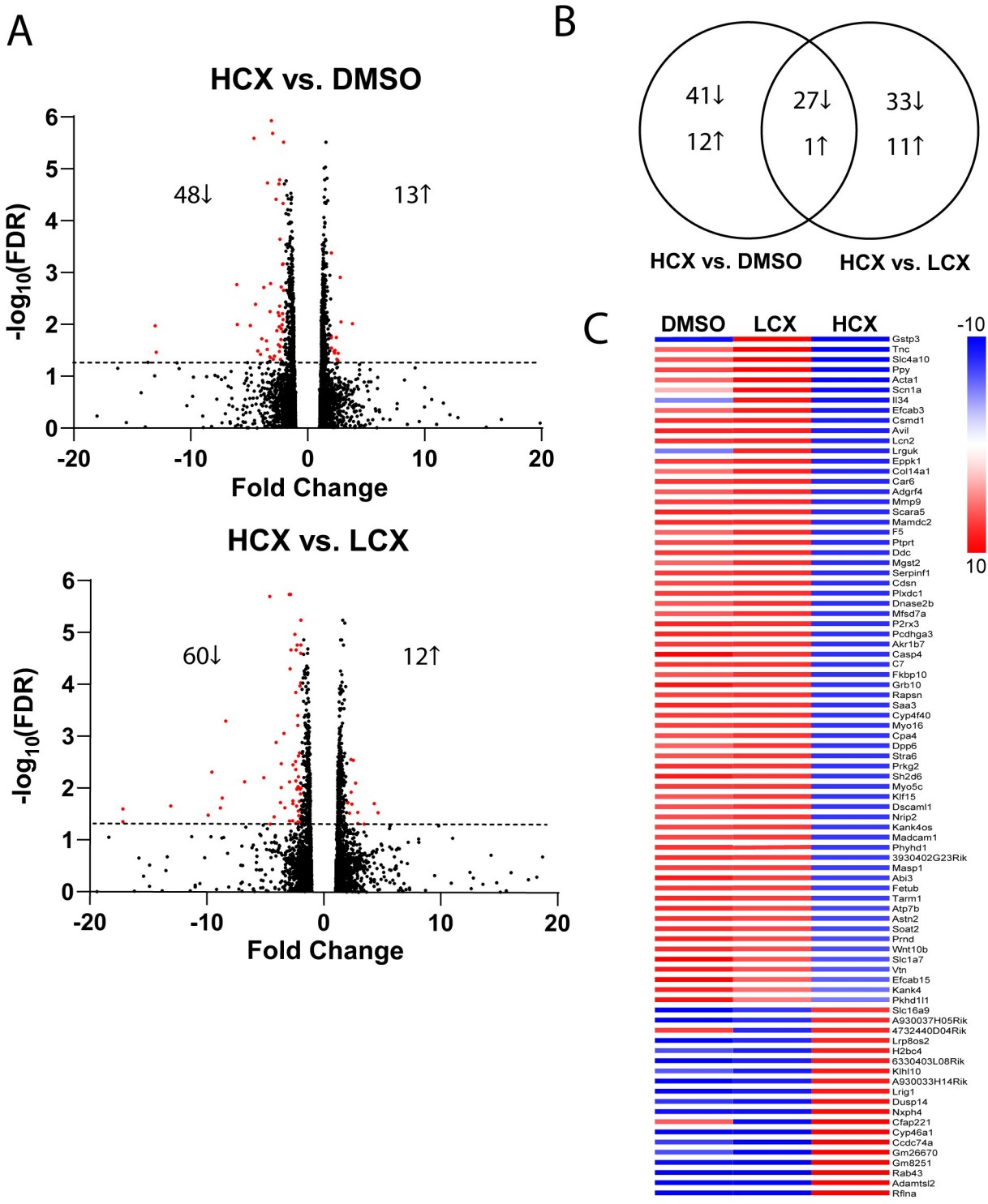

**Fig 6. RNAseq analysis of xyloside-treated Neuro2a cells.** Neuro2a cells were treated with the indicated levels of xylosides for 72h. RNA was extracted and sequenced. A) Volcano plots showing significant changes in gene expression (>50% change, P < 0.5) between HCX and DMSO-treated cells and HCX and LCX-treated cells. B) Venn diagram showing numbers of genes changed with HCX compared to DMSO and LCX treatment. C) Heat map of expressed genes that were changed by xyloside treatment. Color represents deviation from the average expression over all conditions.

genes changed in each condition. Interestingly, majority of genes whose expression was changed were downregulated by exposure to xylosides (Fig 6B). The Venn diagram also indicates that only about 1/3 of the genes were differentially expressed by both HCX vs DMSO and HCX vs LCX. Fig 6C presents a heat map of the level of expression of those genes which are known to be translated into protein as compared to the average level over all conditions. There was no preference in in specific pathways using GO terms. Thus, the changes in cytoskeletal dynamics after LCX treatment are not likely to be due to changes in gene expression.

## Discussion

Xylosides have been used in research as a GAG-biosynthesis inhibitor since the 1970s [14], and continue to be used today. Virtually all published work has used these small molecules in the millimolar range as originally published by Schwartz et al. [24]. Our work reveals a novel action of xylosides on cell morphology that occurs at sub-micromolar concentrations, well below any previously reported to be active. Interestingly, LCX treatment only slightly raised the secretion of CS and HS into the medium, though we did confirm a large increase in GAG secretion by HCX, a major effect of these compounds when used in culture. We did not identify any changes in gene expression in cells exposed to the low concentrations of 4-MU that produced the morphological and cytoskeletal changes, though we did find changes after exposure to HCX.

Several studies have investigated dose-response curves with xylosides on GAG chain production and composition at concentrations in the micromolar range. We found that LCX produced a small change in CS secretion into the medium, with no change in HS, while HCX produced large increases in CS and HS GAG production. In contrast, both LCX and HCX changed GAG chain composition both for CS and HS. For CS, xyloside treatment caused an increase in sulfation of the GalNAc sugars at the 4-position. Our research has found an increase in 4-S sugars to be associated with injury to the CNS [26, 27], and that CS enriched in 4-S GAG acts to reduce neurite outgrowth [26]. These results can be compared to those of Carrino and Caplan [10], who found increased incorporation of $^{35}$S into the medium of chick muscle cultures with concentrations down to 1 μM, with minimal effect on the structure of CS GAG chains, while mM concentrations resulted in GAG chains that were >90% enriched in 6-sulfated GalNAc. Similarly, Weinstein, et al. [28] found increased $^{35}$S secretion into the medium by human chondrocytes treated with 25 μM xylosides, with a maximum attained at 100 μM; higher concentrations then suppressed GAG synthesis. More recently, Persson, et al. [25], systematically looked at GAG chain secretion and composition after treatment with different xylosides in several cell lines. Consistent with other observations, they found an increase in GAG production with 10 μM 4-MU in all cell lines. However, in contrast to our results, they found a decrease in the proportion of ΔHexUA-GalNAc-4S and an increase in the proportion of ΔHexUA-GalNAC-6S in the secreted CS. Because they observed divergent responses in different cell lines, it may be that Neuro2a cells respond differently.

Treatment of Neuro2a cells with HCX, but not LCX, caused an increase in HS secretion into the medium. In contrast, both concentrations of 4-MU caused changes in HS GAG composition. LCX increased ΔHexUA-GlcNS, while HCX further increased the percentage of ΔHexUA-GlcNS, and also increased the percentage of ΔHexUA-2SGlcNS. Knockout of the Ndst1 gene, which controls N-sulfation of HS chains alters many developmental processes, including brain development, in both mice [29] and humans [30]. The knockout mice also have altered responses to both FGF [31] and VEGF [32], which likely contribute to the developmental phenotypes in these animals. It is therefore possible that the increase in ΔHexUA-2SGlcNS could have contributed to the phenomena we observed.

The major cellular change we found with low concentration of 4-MU were in the cytoskeleton. Interestingly, we observed a change in actin dynamics in Neuro2a cells with LCX, but not HCX, while both concentrations changed microtubule dynamics. However, the major phenotype we observed in hippocampal neurons with LCX treatment was enlarged growth cones with microtubule looping. Several previous studies have examined the effect of high concentrations of xylosides on cytoskeleton. Adding 1 mM xyloside to vascular smooth muscle cells lead to a reduction in the number of α-actin containing cytoskeletal filaments [33, 34]. Treatment with 2 mM xyloside treatment impeded tubule formation during nematocyst development, implicating CS in the stabilization of membrane protrusions [35]. However, our observations suggest that there may be specific alterations of the cytoskeleton after exposure to much lower concentrations of xylosides. This suggests that, although LCX has specific morphological actions on both cells types, the underlying mechanism could be different.

Previous research in our lab as well as others has shown that neurons and neuronal cell types are sensitive to GAG chains and changes in sulfation patterns [26, 36, 37]. This sensitivity is linked to several processes that are regulated by cytoskeletal rearrangements such as neural migration, polarity and axon guidance. For example, degradation of GAG chains by enzyme or disruption of biosynthesis by xylosides led to altered neuronal migration [38]. The addition of HS and CS to embryonic rat neurons primarily resulted in neurons with a single long axon. Conversely the addition of DS resulted in neurons with increased dendritic growth that maintained higher levels of microtubule-associated protein 2 expression [39]. In terms of axon guidance, many studies have shown the bifunctionality of HS and CS as guidance cues. HS is commonly associated with permissive substrates while CS, especially 4-sulfated CS, is seen as inhibitory [40]. These actions clearly depend upon alterations in the cytoskeleton.

Because the morphological changes in neurons take several days to develop, we tested the hypothesis that there would be changes in gene expression. However, our RNAseq analysis did not find any RNAs whose level was significantly changed by exposure to LCX as compared to the control (DMSO alone) situation. However, we did find a subset of genes whose expression was significantly increased by HCX over either control or LCX. A large group of these code for proteins in the extracellular matrix, including Tenascin-C, Collagen 14, MMP9 and astrotactin 2, while another group, including α-actin, advillin, protein tyrosine phosphatase receptor type T and ABI family member 3 are involved in controlling cytoskeletal dynamics. However, none of these were changed at concentrations that produced the morphological changes. Treatment of HeLa cells with mM xylosides caused the upregulation of several RNAs related to proteoglycan synthesis [41], but we did not find any overlap with our results. Thus, it is not likely that the morphological changes we observe are due to changes in gene expression.

The question then arises as to what mechanism is mediating these changes in cellular morphology due to LCX treatment. Changes in cytoskeleton are most often induced by changes in cell signaling. Because many signaling pathways are modulated by interactions with cell-surface GAG chains, it is possible that signaling was altered. However, while short-term changes in signaling are mediated through post-translational modifications (PTMs), longer exposure normally produces changes in transcription. As noted above, our RNA-sequencing data shows little to no change in transcription after LCX treatment, indicating that any changes to transcription are not robust and thus alterations in signaling pathways controlling gene expression are not likely the main culprit behind changes in cell morphology. We found that LCX only influenced actin dynamics, while both concentrations caused changes in microtubules. While this might suggest a predominant effect on actin dynamics, it is possible that the larger concentration of secreted GAGs under HCX treatment activated pathways to alter intracellular signaling, antagonizing the effects of LCX.

This leaves changes in PTMs due to xyloside treatment as the most likely driving force behind the altered cytoskeleton. PTMs include phosphorylation, methylation, acetylation and glycosylation. Changes in PTMs are known to alter protein degradation, kinase activity, and intracellular protein localization. This study has shown that both high and low concentration xyloside treatment can alter glycosylation in a dose dependent manner as evidenced by the different disaccharide profiles in the primed GAGs. Previous work has shown that receptors in many signaling pathways are sensitive to changes in GAG sulfation [29]. The differences in changes in glycosylation between LCX and HCX along with the significant increase in CS after HCX treatment which is not found after LCX treatment could explain why the phenomenon is dose dependent.

## Supporting information

**S1 Fig. Microtubule looping in xyloside-treated growth cones.** A) Images of hippocampal neurons treated with either DMSO or LCX. LCX-treated neurons have large growth cones with extensive microtubule looping (arrow). B) Dose response curve for xyloside treatment. Percentage of neurons with looped microtubules at the end of the growth cones increased and peaked at 500 nM xyloside treatment.
(PDF)

**S2 Fig. LCX treatment alters early neurite outgrowth.** Images of hippocampal neurons treated with either DMSO, LCX or HCX 24 h after plating. Arrows point to splayed tubulin at the ends of growing neurites in LCX-treated cultures. Scale bar = 25 μm.
(PDF)

**S3 Fig. Altered Neuro2a morphology in cells treated with LCX.** Cells were transfected with F-tractin (red) and fixed and stained with DAPI (blue) 48 h later. (Left) DMSO-treated Neuro2a cells show typical morphology irregular shape and intense actin staining at the periphery. (Center) LCX-treated cell shows large lamellipodia (arrows) with centripetal actin organization. (Right) HCX-treated cells resemble DMSO-treated cells with irregular shape and peripheral actin staining. Scale = 25 μm.
(PDF)

**S1 Data.**
(XLSX)

## Acknowledgments

Images in this manuscript were acquired in the Light Microscopy Core of the Division of Intramural Research of the National Heart, Lung, and Blood Institute, NIH. We very much appreciate the comments and suggestions of the referee.

## Author Contributions

**Conceptualization:** Caitlin P. Mencio, Yasuhiro Katagiri, Herbert M. Geller.

**Investigation:** Caitlin P. Mencio, Sharada M. Tilve, Masato Suzuki, Kohei Higashi.

**Supervision:** Herbert M. Geller.

**Writing – original draft:** Caitlin P. Mencio.

**Writing – review & editing:** Herbert M. Geller.

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
