## [Decision Letter · Decision Letter 0]

28 Feb 2022

PONE-D-22-03467A Novel Cytoskeletal Action of XylosidesPLOS ONE

Dear Dr. Geller,

Thank you for submitting your manuscript to PLOS ONE. After careful consideration, we feel that it has merit but does not fully meet PLOS ONE’s publication criteria as it currently stands. Therefore, we invite you to submit a revised version of the manuscript that addresses the points raised during the review process.

As requested by both reviewers, the text should be extensively revised, in particular the Methods, because of the lack of many technical details, the Figure legends and the Discussion section. Please submit your revised manuscript by Apr 14 2022 11:59PM. If you will need more time than this to complete your revisions, please reply to this message or contact the journal office at plosone@plos.org. Please include the following items when submitting your revised manuscript:A rebuttal letter that responds to each point raised by the academic editor and reviewer(s). You should upload this letter as a separate file labeled 'Response to Reviewers'.A marked-up copy of your manuscript that highlights changes made to the original version. You should upload this as a separate file labeled 'Revised Manuscript with Track Changes'.An unmarked version of your revised paper without tracked changes. You should upload this as a separate file labeled 'Manuscript'.

We look forward to receiving your revised manuscript.

Kind regards,

Catherine FAIVRE-SARRAILH

Academic Editor

PLOS ONE

Journal Requirements:

"HG; 1ZIAHL006021; NIH"

Reviewers' comments:

Reviewer's Responses to Questions

**Comments to the Author**

1. Is the manuscript technically sound, and do the data support the conclusions?

Reviewer #1: Partly

Reviewer #2: Partly

2. Has the statistical analysis been performed appropriately and rigorously? 

Reviewer #1: Yes

Reviewer #2: No

3. Have the authors made all data underlying the findings in their manuscript fully available?

Reviewer #1: Yes

Reviewer #2: No

4. Is the manuscript presented in an intelligible fashion and written in standard English?

Reviewer #1: Yes

Reviewer #2: Yes

5. Review Comments to the Author

Reviewer #1: The ms by Mencio et al. describes how the cytoskeleton and the morphology of neural cells (here, hippocampal neurons and N2A cells) are affected by nanomolar doses of xylosides, compounds usually applied in the millimolar range to interfere with GAG chain synthesis. The work is principally interesting since it suggests that very subtle changes in GAG chain composition may affect the actin cytoskeleton and thus, cell morphology and migratory behavior. It is also another example of very low doses of a compound exhibiting different effects than high doses.

The reviewer regrets, however, that some findings of the authors seem contradictory, questionable or to the least, unclear for the reader (see below).

One main concern is that the discussion section, while exhaustively reporting on previous work on GAG chain modification by xylosides, tries hard but does not come up with a sort of theory, let alone an explanation for the low dose effect of xylosides described by the authors. Thus, for the reviewer, the conclusion of the study is already contained in the Introduction lines 69-71 ("this study serves as a first step...").

In the Discussion, any attempts to explain the observed effects are limited to "This is an area of research that needs further exploration", repeated several times. "... [low concentration] xyloside treatment may lean to subtle changes in GAG chains having more distinct biological effects" (lines 389-90), repeated just below (lines 406-408): "This could imply that utilizing lower concentrations that result in distinct, but subtle changes to GAG profiles may produce more unique and useful biological outcomes". Besides "more unique and useful" seeming rather "cryptic" terms in biology, these sentences don't explain, but simply repeat the observed findings. The fact that GAGs may interact for ex. with Sema5A (lines 382-386), which in turn may indeed affect the cytoskeleton, doesn't help either because Sema5A is not present here (as far as we know). Still other arguments are simply not new: (lines305-6): "These findings indicate that xyloside treatment can change GAG chain synthesis and composition, and alterations to HS appear to occur in a concentration dependent manner".

Maybe it could have helped if the authors had produced a model/schematic drawing of what they think happens on the molecular level when using low vs. high xyloside concentrations.

Another critical point may be seen in the fact that the authors used the same "LCX" concentration for primary hippocampus cells and for a "neural" cell line (N2A), with which the majority of the work was performed. Many studies have shown that different cell types react differently to a given concentration of xylosides, so it may have been interesting to redo a dose response curve with these cells.

Finally, one may ask whether the N2A cell line used in this study represents really neural cells (neurons), in view of the images shown that are very different from the hippocampal neurons shown in the beginning.

Major points:

Abstract line 32: "... did not alter GAG chain synthesis rates, nor..." the fact that GAG dissaccharide composition was altered is probably crucial and should be mentioned here.

Introduction, line 61: Other studies than the mentioned ref. 16 already used relatively low xyloside concentrations, for ex. Carrino & Caplan 1994 who used 1µM, which is not that far from 500nM (in comparison to mM).

Materials and Methods: for the reviewer, at least parts of this section are lacking some crucial details.

Controls for xyloside treatment are always termed "DMSO"; however, we do not know how much DMSO was introduced into the cultures, as we do not learn at what stock concentration xyloside was used. Other, similar studies always stated that they used the highest DMSO concentration for controls (corresponding to the highest xyloside conc. applied). So the reader may ask if DMSO addition already modifies something?

Cell culture line 83: cell density 8-10k/well (is this 12-well, 24-well, ...?).

24mM KCl was added to the culture of hippocampal neurons, why ? (usually this is done to depolarize the neurons, but here?).

Line 146- Neurite otgrowth and growth cone analysis: "both longest and total neurite measurements...": "total neurite" doesn't appear anywhere in the ms, and does it mean total length of all neurites ? How were neurite length, and the growth cone area exactly measured (ImageJ plugin?).

Cytoskeleton dynamics: EB3-GFP and Ftractin-mCherry transfections are not detailed (maybe a recent reference would suffice). Also, the reviewer asks whether, since transfections were performed after exposure to xyloside, are cells reacting the same to the transfections as without xyloside, what is the percentage of transfected cells, does transfection itself have other consequences on cell morphology ?

Results:

Figure 1: An overview image showing more than one neuron would be welcome; here, we just see one well spread growth cone (GC) formed by an LCX treated neuron, while the GC from control and DCX neurons appear rather collapsed. Also, the LCX neurite is much shorter than the other two, so I guess we are dealing with somewhat extreme (so called "typical") examples.

Supp Figure 1: to me, this is the "key" figure of the ms, as it shows the dose response curve for xyloside treatment, on which the rest of the ms, including the work on N2A, is based. The reviewer is puzzled when looking at the images: how many neurons do we see, particularly in the right LCX image? What looks like GC, contains also nuclei? Then, how were "looping MTs" quantified? In a "collapsed" GC like those seen on control neurons (left image) they would be hard to count. This is also not specified in M&M.

Supp Figure 2 is the basis for N2A cell line-based work. (Line 217) "... LCX treated N2A exhibit actin-rich lamellipodia, not observed in DMSO or HCX treated N2A". Besides the fact that in the LCX image the red color appears more saturated to me than in the other two images (see the completely flat yellow zones); this finding appears completely contradictory to Fig. 3 (see below).

Figure 2: Looking at the images, it seems strange to me that at t=0, the tracked cells do not show any extensions, which then develop at t=80 and t=160 min where cells really look like migratory cells (as if someone had given the "start signal"), although cells were kept under the same conditions already for 48h.

Figure 3: In the figure legend it says "phalloidin staining on fixed cells", in the text it says Ftractin mCherry transfection (normally used for live viewing?), what in fact do we see here ?

(lines 243-250) LCX treated N2A lamellipodia had significantly fewer actin bundles...; additionally, ...less robust actin bundles in LCX lammellipodia. How do the authors explain that in Supp Fig 2, DMSO or HCX treated N2A did not exhibit actin-rich lamellipodia, while here they state in some sort the opposite? Moreover, the cells in Fig. 3 do not resemble at all those shown in Supp Figure 2, let alone the (hippocampal) real neurons in Fig. 1. Here and elsewhere, figure legends (M&M) are not precise enough (different culture times between Supp Fig. 2 and Fig. 3, for example?).

To me, the HCX cell looks polarized, much more than the LCX cell; this is even more obvious in Figure 4A.

Figure 4: "These results [...] suggest LCX treatment alters cytoskeleton dynamics, which lead to changes in morphology and movement"; yes, but in Fig. 4 (EB3 comet movement) there is no difference between LCX and HCX treated N2A cells, only between DMSO and xyloside treatment. This is somehow in contrast to the assumption that low, but not high xyloside concentrations lead to major morphology changes (see e.g. Introduction line 64-65; Figs. 1, 2, Supp Fig2; discussion line 405-406: "no visible effect is observed in 1mM xyloside treated neurons or neuro2A cells").

Besides, as in Fig. 3, it looks to me that the cells presented differ in polarity, the HCX cell being the most polarized.

Figure 5: the legend should at least mention what means 0S, NS, 6S, 2SNS, especially for the broad audience of PlosOne who may not all be familiar with these terms.

The figure shows that CS GAG concentration in the medium is sort of xyloside dose dependent, whereas both low and high xyloside doses provoke the same change in CS disaccharide composition. HS GAG concentration in the medium is not altered by LCX, but the relative abundance of N-sulfated GAGs is. This seems interesting to me with regard to the role of N-sulfated HS reported in the literature (see e.g. Grobe et al 2005, Development 132: 3777), and this point could well be analyzed a little more in the Discussion section.

Another point: the difference between LCX effects on CS and HS GAG production may in fact be due to the DMSO controls already producing some HS (~0.1 ng/µL), but no CS.

The problem I see here, is that we do not learn whether the small effect of LCX on CS GAG concentration could have an effect on cell morphology, or whether the altered sulfation of HS GAGs does. These questions may partly be answered by, for example, rising (exogenous) CS concentration in the culture medium (i.e. not through xyloside treatment), or by selectively suppressing N-sulfation.

Figure 6C: there seem to be at least some genes whose expression is changed between DMSO and LCX ? The formulation (lines 314-15) "we found no genes whose change in expression was different between LCX and DMSO..." is not very clear.

Discussion:

When reading the discussion, two more points came to my mind: i) is the observed low dose effect on cell morphology specific for "neural" cells, or may it affect also the other cell types (muscle etc.) that had already been used in xyloside experiments? ii) It may have been interesting to include HS/CS staining of the cells after LCX/HCX treatment, as seen for ex. in the Nishimura et al. work (ref. 20).

Minor points:

The abbreviations for HS, CS (and KS) need be introduced (Introduction line 39).

Figure legends are often too short. Ex., Figure1: "altered cytoskeleton", the label is not indicated (tubulin?). Figure 2, "letters" should rather read "numbers". Figures 3 and 4, see before. Also in Fig.4, "distribution plots of speed, persistence and comets" ("lifetime" missing?), and in the histograms themselves is marked "200715_Speed" etc. which doesn't tell us anything. Figure 5, add abbreviations.

Supp Fig. 1, the bar is missing.

Some typo etc. errors, such as "DMOS" (line 104), "in in" (line 323), "on at" (line 344), "1C" (1µ?; line 346), "the change...were..." (line 356).

Reviewer #2: PONE-D-22-03467

This paper described that a nM range concentration of xyloside (4-methyl-umbelliferyl-�-D-xylopyranoside) had effects in cellular morphology of primary neurons and Neuro2A cells, while a mM range does not. Authors report changes in growth cone size with an increase in microtubules looping in primary neuronal cultures, reduced cell migration and altered actin bundling in Neuro2A cell line after 48h in low concentration of xyloside (LCX). Even though these observations have not been made previously, the data does not offer a mechanistic explanation for the observations, nor it does establish a clear link between the morphological changes and changes in proteoglycans (PGs) and/or glycosaminoglycans (GAGs) synthesis.

Evaluation of this paper has been difficult because of a lack of some technical details, incongruent description of the data in figure legends and text and a somewhat limited discussion of the data.

Major points:

- Xyloside treatment impairs the incorporation of GAGs in the PGs core proteins and increases the secretion of free GAG chains into the media. Thus, these changes in morphology could be due to partially modified core proteins at LCX conditions, and analyzing changes in GAG composition of cell and matrix associated proteoglycans could be important to evaluate here. Alternatively, if the partially glycosylated core protein are responsible, similar morphological changes could be obtained by knocking-down expression of Xylosyltransferase 1 and 2 in these cells.

- In the discussion the possibility that an intracellular function for GAGs should also be discussed considering the recent publication of Fang et al. (https://doi.org/10.1016/j.immuni.2021.03.011), where he found the participation of GAGs chains in polymerization and activation of STING at the Golgi level in immune cells. Still undiscovered intracellular functions for GAGs in neuronal cells could be responsible for the morphological changes described here.

-Line 48. Please clarify that only CS/DS and HS have the common Xyl-Gal-Gal linkage to core protein. KSPGs are bound to PGs core proteins by N- or O- linkage sugars and as such are not influenced by xyloside treatment.

- Line 96. Please clarify the “normal cell culture conditions”; Is this with or without FBS? This is important to evaluate your GAGs composition results.

- Line 114. Methods indicates that a 30K filter was used to separate CS disaccharides from HS chains. Could this be a typo? HS chains will go through a 30K filter.

- Line 241-255. Text accompanying Figure 3 explained an F-tractin-mCherry experiment while the figure 3 legend described a phalloidin staining experiment. Which one is it? Or the wrong figure was included?

- Line 269. It is unclear how many cells were evaluated for microtubules dynamics in Figure 4B.

- Line 285. It is unclear the number of samples quantified per treatment group and the statistical analysis used to assess significance (see line 283).

- Line 328 and 176. Significance stated in Materials and methods is different that the one stated in the figure legend. Data appear to be adjusted by false discovery rate (FDR) but no threshold was stated in the figure legend. Also, the genes listed in figure 6C are impossible to read. A data file with the list of genes and fold changes should be supplied as supplemental data. Please explain the -10 to 10 color scale used.

- Raw and processed RNA-seq data should be make available in the National Center for Biotechnology Information Gene Expression Omnibus.

Minor points:

- Antibodies used should be specified in Materials and Methods.

- Line 66. Xiloside is misspelled.

- Line 104. Typo, it should read DMSO.

- Line 171. First mention of LCX and HCX, please define here.

- Line 181 and 210. Punctuation should be corrected.

- Line 249. There is no Figure 2C so this probably should read Figure 3C.

- Line 346. …with concentrations down to 1 C? Unclear, Typo?

- Line 378. Defined DS.

6. PLOS authors have the option to publish the peer review history of their article (what does this mean?). If published, this will include your full peer review and any attached files.

Reviewer #1: No

Reviewer #2: No

---

## [Author Response · Author response to Decision Letter 0]

1 Apr 2022

Response to Reviewer :

We thank the reviewer for their thorough and incredibly valuable comments and suggestions on the manuscript. We appreciate that this is a novel observation and have tried to answer each of the reviewers’ comments with appropriate changes in either the text or the figures. We have organized this response such that each comment or suggestion has a reply in different font from the comment. We hope that this submission is suitable for publication.

Herbert M. Geller, for the authors.

The ms by Mencio et al. describes how the cytoskeleton and the morphology of neural cells (here, hippocampal neurons and N2A cells) are affected by nanomolar doses of xylosides, compounds usually applied in the millimolar range to interfere with GAG chain synthesis. The work is principally interesting since it suggests that very subtle changes in GAG chain composition may affect the actin cytoskeleton and thus, cell morphology and migratory behavior. It is also another example of very low doses of a compound exhibiting different effects than high doses.

The reviewer regrets, however, that some findings of the authors seem contradictory, questionable or to the least, unclear for the reader (see below).

One main concern is that the discussion section, while exhaustively reporting on previous work on GAG chain modification by xylosides, tries hard but does not come up with a sort of theory, let alone an explanation for the low dose effect of xylosides described by the authors. Thus, for the reviewer, the conclusion of the study is already contained in the Introduction lines 69-71 ("this study serves as a first step..."). 

In the Discussion, any attempts to explain the observed effects are limited to "This is an area of research that needs further exploration", repeated several times. "... [low concentration] xyloside treatment may lean to subtle changes in GAG chains having more distinct biological effects" (lines 389-90), repeated just below (lines 406-408): "This could imply that utilizing lower concentrations that result in distinct, but subtle changes to GAG profiles may produce more unique and useful biological outcomes". Besides "more unique and useful" seeming rather "cryptic" terms in biology, these sentences don't explain, but simply repeat the observed findings. The fact that GAGs may interact for ex. with Sema5A (lines 382-386), which in turn may indeed affect the cytoskeleton, doesn't help either because Sema5A is not present here (as far as we know). Still other arguments are simply not new: (lines305-6): "These findings indicate that xyloside treatment can change GAG chain synthesis and composition, and alterations to HS appear to occur in a concentration dependent manner". 

Maybe it could have helped if the authors had produced a model/schematic drawing of what they think happens on the molecular level when using low vs. high xyloside concentrations.

We agree that the discussion was not as informative as we would all like. Unfortunately, we could not conduct the mechanistic studies that would be important to answer this question, and so any interprétations we have are speculative. We did not include a model/schematic, as we think that any figure would be based on these speculations, and a reader who simply looked at figures (as many do) could be misled. Instead, we have replaced the verbiage cited by the reviewer with a more focussed discussion about what our data do support and do not. 

Another critical point may be seen in the fact that the authors used the same "LCX" concentration for primary hippocampus cells and for a "neural" cell line (N2A), with which the majority of the work was performed. Many studies have shown that different cell types react differently to a given concentration of xylosides, so it may have been interesting to redo a dose response curve with these cells.

While we began these studies with hippocampal neurons, we realized that certain data on biochemistry would be difficult to determine using primary cultures which can be highly variable. For consisency, we therefore evaluated the two concentrations of xylosides that were used in hippocampal neurons on Neuro2a cells and found that they had differential concentration-dependent effects at the same concentrations used in neurons, and therefore continued with these concentrations. 

Finally, one may ask whether the N2A cell line used in this study represents really neural cells (neurons), in view of the images shown that are very different from the hippocampal neurons shown in the beginning. 

Neuro2a cells are mouse neuroblastoma cells and have been used in many different settings to examine signal transduction and biochemistry. Because they are a tumor cell line, they may exhibit various morphologies depending upon treatment, as we show here. There are many published papers that present data from both hippocampal neurons and Neuro2A cells, but we do not presume that all measurements will be the same in both. 

Major points: 

Abstract line 32: "... did not alter GAG chain synthesis rates, nor..." the fact that GAG dissaccharide composition was altered is probably crucial and should be mentioned here.

We agree and have modified the manuscript both in the introduction and discussion to emphasize these results.

Introduction, line 61: Other studies than the mentioned ref. 16 already used relatively low xyloside concentrations, for ex. Carrino & Caplan 1994 who used 1µM, which is not that far from 500nM (in comparison to mM).

We did cite this paper and its results in the discussion, but we have now included it in the introduction as well.

Materials and Methods: for the reviewer, at least parts of this section are lacking some crucial details. 

Controls for xyloside treatment are always termed "DMSO"; however, we do not know how much DMSO was introduced into the cultures, as we do not learn at what stock concentration xyloside was used. Other, similar studies always stated that they used the highest DMSO concentration for controls (corresponding to the highest xyloside conc. applied). So the reader may ask if DMSO addition already modifies something?

We have edited the methods to show the stock concentrations and the dilution factor for DMSO and xylosides used in the paper. Additionally, we had compared DMSO treatment to untreated cells and saw no noticable difference in morphology. No previous studies have shown any changes in the cell at the percentage of DMSO we utilize.

Cell culture line 83: cell density 8-10k/well (is this 12-well, 24-well, ...?). 

We have clarified which cell culture dishes were used as well as cell densities for each type of dish.

24mM KCl was added to the culture of hippocampal neurons, why ? (usually this is done to depolarize the neurons, but here?).

This concentration of KCl added to neurobasal medium has been found to promote neuronal health and survival of cultured neurons, and we do this in our lab protocols (Pearson CS, Mencio CP, Barber AC, Martin KR, Geller HM. Identification of a critical sulfation in chondroitin that inhibits axonal regeneration. Elife. 2018 May 15;7:e37139. doi: 10.7554/eLife.37139. PMID: 29762123; PMCID: PMC5976435.) A literaturer search shows that there are many other publications using high KCl to promote neuronal culture survival. 

Line 146- Neurite outgrowth and growth cone analysis: "both longest and total neurite measurements...": "total neurite" doesn't appear anywhere in the ms, and does it mean total length of all neurites ? How were neurite length, and the growth cone area exactly measured (ImageJ plugin?).

We have added more detail into the materials and methods section on the analysis and data collection for neurites and growth cones. 

Cytoskeleton dynamics: EB3-GFP and Ftractin-mCherry transfections are not detailed (maybe a recent reference would suffice). Also, the reviewer asks whether, since transfections were performed after exposure to xyloside, are cells reacting the same to the transfections as without xyloside, what is the percentage of transfected cells, does transfection itself have other consequences on cell morphology ?

The tranfection order was determined based on optimization of both cell survival and transfection efficiency. We found routinely about 40% of cells were transfected. We have used this transfection protocol to express GFP (without xylosides) in other experiments with Neuro2A cells (Agbaegbu Iweka C, Hussein RK, Yu P, Katagiri Y, Geller HM. The lipid phosphatase-like protein PLPPR1 associates with RhoGDI1 to modulate RhoA activation in response to axon growth inhibitory molecules. J Neurochem. 2021 May;157(3):494-507. doi: 10.1111/jnc.15271. Epub 2021 Jan 3. PMID: 33320336; PMCID: PMC8106640) (now cited in the manuscript) and have not observed changes in morphology. 

Results:

Figure 1: An overview image showing more than one neuron would be welcome; here, we just see one well spread growth cone (GC) formed by an LCX treated neuron, while the GC from control and DCX neurons appear rather collapsed. Also, the LCX neurite is much shorter than the other two, so I guess we are dealing with somewhat extreme (so called "typical") examples.

We have added images of several HC neurons at low power. Unfortunately, it is difficult to display the large growth cônes and looping at that magnification, so we have retained the higher res images.

Supp Figure 1: to me, this is the "key" figure of the ms, as it shows the dose response curve for xyloside treatment, on which the rest of the ms, including the work on N2A, is based. The reviewer is puzzled when looking at the images: how many neurons do we see, particularly in the right LCX image? What looks like GC, contains also nuclei? 

Indeed, there were several neurons in each image as indicated by the DAPI staining, making it difficult to interpret. We have therefore replaced image in the figure with one of a single neuron with the representative phenotype. We hope that these images are more easily interpreted.

Then, how were "looping MTs" quantified? In a "collapsed" GC like those seen on control neurons (left image) they would be hard to count. This is also not specified in M&M.

 We have edited our materials and methods to provide clarity on how we determined the presence of looped MTs.

Supp Figure 2 is the basis for N2A cell line-based work. (Line 217) "... LCX treated N2A exhibit actin-rich lamellipodia, not observed in DMSO or HCX treated N2A". Besides the fact that in the LCX image the red color appears more saturated to me than in the other two images (see the completely flat yellow zones); this finding appears completely contradictory to Fig. 3 (see below).

We appreciate this comment. N2A cells have various morphologies and our goal was to select representative images from cells grown at the same time, and so the cells in Supp. Fig. 2 were somewhat different from those in Fig. 3. We have now selected images of f-tractin transfected cells for Supp. Fig. 2 which are contemporaneous with those in Fig. 3. These images more clearly display the morphologies quantified in N2A cells. 

Figure 2: Looking at the images, it seems strange to me that at t=0, the tracked cells do not show any extensions, which then develop at t=80 and t=160 min where cells really look like migratory cells (as if someone had given the "start signal"), although cells were kept under the same conditions already for 48h. 

These images are included as examples of how we can track movement. Not all the cells are migrating all the time. However, we have replaced the former images with images showing the various morphologies. 

Figure 3: In the figure legend it says "phalloidin staining on fixed cells", in the text it says Ftractin mCherry transfection (normally used for live viewing?), what in fact do we see here ?

We thank the referee for pointing this out. We have corrected the text to correspond with the figure.

(lines 243-250) LCX treated N2A lamellipodia had significantly fewer actin bundles...; additionally, ...less robust actin bundles in LCX lammellipodia. How do the authors explain that in Supp Fig 2, DMSO or HCX treated N2A did not exhibit actin-rich lamellipodia, while here they state in some sort the opposite? Moreover, the cells in Fig. 3 do not resemble at all those shown in Supp Figure 2, let alone the (hippocampal) real neurons in Fig. 1. Here and elsewhere, figure legends (M&M) are not precise enough (different culture times between Supp Fig. 2 and Fig. 3, for example?).

We hope that we have answered this comment with the images in the new Supp. Fig. 2 which clearly shows the actin bundles in the LCX-treated cells. We also provide more details on the culture and transfection time in the figure legends. 

To me, the HCX cell looks polarized, much more than the LCX cell; this is even more obvious in Figure 4A.

 We did not observe consistent changes in polarity. We have selected new representative images that show cells in a more similar state of polarization across all three experimental conditions.

Figure 4: "These results [...] suggest LCX treatment alters cytoskeleton dynamics, which lead to changes in morphology and movement"; yes, but in Fig. 4 (EB3 comet movement) there is no difference between LCX and HCX treated N2A cells, only between DMSO and xyloside treatment. This is somehow in contrast to the assumption that low, but not high xyloside concentrations lead to major morphology changes (see e.g. Introduction line 64-65; Figs. 1, 2, Supp Fig2; discussion line 405-406: "no visible effect is observed in 1mM xyloside treated neurons or neuro2A cells").

Besides, as in Fig. 3, it looks to me that the cells presented differ in polarity, the HCX cell being the most polarized.

 Thank you for pointing out this issue. We have now modified the text in both the results and discussion to point out the difference in the response of the actin and tubulin cytosekeletons to xylosides, i.e., that both concentrations alter MT dynamics, but only LCX alters actin. 

Figure 5: the legend should at least mention what means 0S, NS, 6S, 2SNS, especially for the broad audience of PlosOne who may not all be familiar with these terms. 

The figure shows that CS GAG concentration in the medium is sort of xyloside dose dependent, whereas both low and high xyloside doses provoke the same change in CS disaccharide composition. HS GAG concentration in the medium is not altered by LCX, but the relative abundance of N-sulfated GAGs is. This seems interesting to me with regard to the role of N-sulfated HS reported in the literature (see e.g. Grobe et al 2005, Development 132: 3777), and this point could well be analyzed a little more in the Discussion section. 

We agree that the these findings were not adequately discussed. We have altered the discussion of the GAG sulfation to also indicate biological implications We have also included the definitions in the figure legend as requested.

Another point: the difference between LCX effects on CS and HS GAG production may in fact be due to the DMSO controls already producing some HS (~0.1 ng/µL), but no CS.

The problem I see here, is that we do not learn whether the small effect of LCX on CS GAG concentration could have an effect on cell morphology, or whether the altered sulfation of HS GAGs does. These questions may partly be answered by, for example, rising (exogenous) CS concentration in the culture medium (i.e. not through xyloside treatment), or by selectively suppressing N-sulfation.

We agree that these are possibilities. Unfortunately, we are unable to conduct these experiments within a reasonable time frame, as the NIH has been quite strict about lab occupancy. We did think about adding the conditioned medium from HCX cells to the LCX condition, but this would still be contaminated by xylosides. In response to this and the suggestion above, we have included these possibilities in the discussion for potential future experiments. 

Figure 6C: there seem to be at least some genes whose expression is changed between DMSO and LCX ? The formulation (lines 314-15) "we found no genes whose change in expression was different between LCX and DMSO..." is not very clear.

According to our analysis using the Partek Flow pipeline, there were no genes whose expression was significantly changed (FDR < 0.05) between the LCX and DMSO conditions. Fig. 6C included all genes whose expression was significantly different between either LCX and HCX or DMSO and HCX, but we put in the relative change in expression for all conditions. Thus, the top one, Gstp3, was only signficantly changed between HCX and LCX. 

Discussion:

When reading the discussion, two more points came to my mind: i) is the observed low dose effect on cell morphology specific for "neural" cells, or may it affect also the other cell types (muscle etc.) that had already been used in xyloside experiments? ii) It may have been interesting to include HS/CS staining of the cells after LCX/HCX treatment, as seen for ex. in the Nishimura et al. work (ref. 20).

 It is clearly possible that other cell types may respond similarly. As noted in the manuscript, few papers used comparable concentrations of xylosides in their experiments, so there is no point of reference. As to the staining with HS/CS, be believe that the biochemical experiments are more informative, as they looked at both production and composition which is not possible with immunocytochemistry.

Minor points:

The abbreviations for HS, CS (and KS) need be introduced (Introduction line 39).

 We have introduced these abbreviations

Figure legends are often too short. Ex., Figure1: "altered cytoskeleton", the label is not indicated (tubulin?). 

 Thanks, we have now indicated that this is tubulin staining, and added additional text to the other legends as well. 

Figure 2, "letters" should rather read "numbers". 

 Thanks again. Corrected.

Figures 3 and 4, see before. Also in Fig.4, "distribution plots of speed, persistence and comets" ("lifetime" missing?), and in the histograms themselves is marked "200715_Speed" etc. which doesn't tell us anything. 

 We thank the reviewer for catching these errors. They have been corrected.

Figure 5, add abbreviations. 

 Done

Supp Fig. 1, the bar is missing. FIxed

Some typo etc. errors, such as "DMOS" (line 104), "in in" (line 323), "on at" (line 344), "1C" (1µ?; line 346), "the change...were..." (line 356).

 Thank you. We hope this version is free of typos.

---

## [Decision Letter · Decision Letter 1]

28 Apr 2022

PONE-D-22-03467R1A Novel Cytoskeletal Action of XylosidesPLOS ONE

Dear Dr. Geller,

Thank you for submitting your manuscript to PLOS ONE. After careful consideration, we feel that it has merit but does not fully meet PLOS ONE’s publication criteria as it currently stands. Therefore, we invite you to submit a revised version of the manuscript that addresses the points raised during the review process.

First, there is a problem with the PDF of the revised manuscript, which does not include anymore Figure 3 and Figure 6 !!! Since the legends of these two Figs are still present in the text, I assume that reviewer 1 took into account Fig 3 and Fig 6, whereas reviewer 2 found the revised version very problematic.Please, take into account all the precise comments of reviewer 1 including: 1-adding HCX image in Supp Fig.1, 2-adding for Fig. 2  the total distance for cell migration and significance values, please read carefully the text and correct the typos. As mentioned by Reviewer 2, you are indicating in the point-by-point answer: "we have added images of several HC neurons at low power”, no changes were made to Figure 1, is it shown in a suppl Fig ?Both reviewers acknowledge that the revised manuscript has been reworked and improved and we hope that you could now submit a final revised version of the manuscript.

We look forward to receiving your revised manuscript.

Kind regards,

Catherine FAIVRE-SARRAILH

Academic Editor

PLOS ONE

Journal Requirements:

Reviewers' comments:

Reviewer's Responses to Questions

**Comments to the Author**

1. If the authors have adequately addressed your comments raised in a previous round of review and you feel that this manuscript is now acceptable for publication, you may indicate that here to bypass the “Comments to the Author” section, enter your conflict of interest statement in the “Confidential to Editor” section, and submit your "Accept" recommendation.

Reviewer #1: (No Response)

Reviewer #2: (No Response)

2. Is the manuscript technically sound, and do the data support the conclusions?

Reviewer #1: Yes

Reviewer #2: Partly

3. Has the statistical analysis been performed appropriately and rigorously? 

Reviewer #1: Yes

Reviewer #2: Yes

4. Have the authors made all data underlying the findings in their manuscript fully available?

Reviewer #1: Yes

Reviewer #2: Yes

5. Is the manuscript presented in an intelligible fashion and written in standard English?

Reviewer #1: Yes

Reviewer #2: Yes

6. Review Comments to the Author

Reviewer #1: also uploaded as PDF file.

The reviewer acknowledges that the revised ms has been profoundly reworked and greatly improved, although I am still not completely convinced by all arguments provided.

- I agree that the GAG chain synthesis and composition studies, and transfections, are easier to perform with N2A cells, but for the rest I personally do not see why hippocampal (or cortical? or maybe even adult DRG) neuron cultures are "too variable" as you say. Since your group is working on axon guidance, regeneration, etc., and you show here at the beginning of your ms how low dose xyloside treatment affects MT looping, growth cone stalling and neuronal morphology, I wonder how you will be able to "translate" results from your in vitro work with N2A cells lacking growth cones and even morphologically not really resembling neurons, for use in your other studies. Sure this is a personal "regret" of the reviewer, not relevant for publication in PLOSOne.

- You say that for Fig.1 you added images of several neurons at low power, but those are not found in the revised ms? Could be a Supplemental Figure.

- Supp Fig.2: Overall, there is now more similarity with Fig.3. However, the caption on the fig. does not correspond to the legend in the ms (line 584). At the same time, the caption on the fig. describes the facts better than the ms legend ("increased levels of interior phalloidin actin staining" vs. "increased levels of actin"; "actin" should at least read F-actin by the way). I think the legend to Supp Fig.2 should be more precise to not induce us into erroneous interpretation of Fig.3, which has obviously not changed (?).

- The discussion has been indeed improved. It does however not deal with the question where the effect on the actin cytoskeleton of LCX vs. HCX treatment may attack. Could there be an effect already during GAG synthesis in Golgi/cytosol and subsequent PG transport, or is there only an effect on (extracellular) signaling via secreted or membrane-bound PGs? (Let's say that is a question that would interest me personally, but you need not answer it).

- There are still some errors and typos, see below.

Line numbers refer to the Word (.docx) document "Final Revision" !

Abstract: the reformulated lines 29-31 (-32) are not very "elegant", and the end may be misleading ("higher concentrations had minor effects"). I'd propose something like:

To our surprise, we found that concentrations of xylosides in the nanomolar to micromolar range had major effects on cell morphology of hippocampal neurons as well as of Neuro2a cells, affecting both actin and tubulin cytoskeletal dynamics. Such effects/morphological changes were not observed with higher xyloside concentrations.

Line32: Xylosides ... produces...

Line33: ...large change in GAG chain synthesis rate

Finally, you did not include the effect on GAG composition in the Abstract, why? You have done so in the Introduction, where you state that your study may contribute to understanding "how a minor shift in GAG composition can affect biological processes..."

Cell culture: Stock solutions are now described, but it would not have been necessary to do it two times (lines 93, 107).

Line108: 10 m should probably read 10 min.

Growth cone analysis:

Lines164-167: this is a bit unclear for the reader. "The randomized files were then numbered sequentially and saved for reference. Duplicates of these files had all identifying information removed and then the numbered files were analyzed."

I guess that the duplicate files without information were analyzed (to make for 'double blind')? Here it sounds as if the reference files (the "numbered" ones) were analyzed. I'd prefer a simple: "Analysis was then performed on duplicates of these files from which all identifying information had been removed".

Lines162- : this does not really answer my question how "collapsed" GC were counted? (as seen in Fig.1: LCX shows a neat, large GC, DMSO an almost collapsed, and HCX no visible GC at all, making it impossible to count/evaluate microtubules).

Results:

Not very "elegant" beginning: "we sought to..., but we sought to..."; and the first sentence is not really true since you did not want to inhibit GAG synthesis here. Maybe you could start with something like "Previous studies on GAG chain synthesis had used...

Here, we wanted to establish a dose-response curve... to determine...".

Fig.1: You could have at least added F-actin staining of those "real" neurons since the rest of the paper is mostly about lamellae and F-actin on N2A cells. F-actin is shown in Supp Fig.1, but there an HCX image is missing.

Fig.2: As in the text you say that velocity and total distance were significantly different, this should be shown in the figure (that shows only velocity), or at least the significance values for total distance mentioned in the text.

Fig.3: the image selected in Supp Fig.2 is a bit closer in comparison now, but I'm still not convinced: what exactly do you designate lamellipodium here in Fig.3 (clearly identifiable in the Supp. Fig.2 for the LCX cell). We should see (if I get it right??) that in LCX treated cells there are well-formed lamellipodia (reminiscent of neural growth cone), but less and thinner actin bundles than in HCX cells.

Several typos in Supp Fig.2 legend on the figure itself (but not in the manuscript).

Line 273: (Figure 2C) should read Fig. 3C.

Discussion:

Lines 303-4: "Treatment with xyloside treatment..."

Line 408: I don't see how you can suggest a different action of LCX on hippocampal neurons and N2A cells based on MT looping in growth cones, since the latter don't form a growth cone (at least not in your study).

Line 442: ...caused changed...

Line 443: predominant effect of actin, or on actin?

Line 452: full stop missing.

Line 454: "...the phenomenology is dose dependent". Normally, phenomenology is a science (sort of) and cannot be dose dependent.

Reviewer #2: PONE-D-22-03467R1

Even though the paper’s text and figures have been extensively changed, the lack of further crucial experiments to clarify the function of CS/HS in this phenomenon is disappointing. As suggested previously by the reviewers, this paper needs additional experimental approaches before it is ready for publication.

Furthermore, changes stated by the authors were not made. For example for Figure 1 “we have added images of several HC neurons at low power”, but no changes were made to Figure 1. Also, the new version does not include Figure 3 or 6. In particular for Figure 3 since the figure legend changed substantially, it is unclear if the figure did too.

As for the swapping of images in supplemental Figure 2 to match the look of Figure 3, it is problematic to me. The cell morphology is so radically different that I wonder what the authors consider to be a representative image. Now the cells in new Figure 2A looks different than cells in supplemental Figure 2 and original Figure 3. Why? A suitable explanation should be offered to the readers in particular when the whole paper is based on cytoskeleton differences between cell treatments.

7. PLOS authors have the option to publish the peer review history of their article (what does this mean?). If published, this will include your full peer review and any attached files.

Reviewer #1: No

Reviewer #2: No

---

## [Author Response · Author response to Decision Letter 1]

26 May 2022

We really appreciate the time the reviewer has taken with this manuscript. We have made additional changes in response to the comments and questions, and have attempted to answer each of the points made by the referee. Because we consider these comments to have greatly improved the manuscript, we have added an acknowledgment of the referee. We hope that these changes have made the manuscript acceptable for publication in PLOS One.

The reviewer acknowledges that the revised ms has been profoundly reworked and

greatly improved, although I am still not completely convinced by all arguments

provided.

- I agree that the GAG chain synthesis and composition studies, and transfections, are

easier to perform with N2A cells, but for the rest I personally do not see why

hippocampal (or cortical? or maybe even adult DRG) neuron cultures are "too variable"

as you say. Since your group is working on axon guidance, regeneration, etc., and you

show here at the beginning of your ms how low dose xyloside treatment affects MT

looping, growth cone stalling and neuronal morphology, I wonder how you will be able

to "translate" results from your in vitro work with N2A cells lacking growth cones and

even morphologically not really resembling neurons, for use in your other studies. Sure

this is a personal "regret" of the reviewer, not relevant for publication in PLOSOne.

I would not argue that the questions you raise are important, just that we don’t have the resources to address them at this time. While the cells are quite different, the result that low concentrations of xylosides have specific actions on the cytoskeleton is likely to be a general effect on many different cell types (though we have not looked a many).

- You say that for Fig.1 you added images of several neurons at low power, but those are

not found in the revised ms? Could be a Supplemental Figure.

Unfortunately, low power images of 72-h treated neurons do not have adequate resolution for fine tubulin or actin filaments, as we plated at low density to avoid neurite overgrowth. However, we have added a supplementary figure showing that differences in neurite morphology could be observed as early as 24 h after plating. Because these cells had much shorter neurites, we are able to present images with more than one cell at a time at a lower resolution than in the other figures. I am also including a low power reviewers’ figure which does not have the adequate resolution to demonstrate the microtubule morphology, but does show general morphology. We hope that these additional images will help interpret the phenomenon. 

- Supp Fig.2: Overall, there is now more similarity with Fig.3. However, the caption on

the fig. does not correspond to the legend in the ms (line 584). 

At the same time, the caption on the fig. describes the facts better than the ms legend ("increased levels of interior phalloidin actin staining" vs. "increased levels of actin"; "actin" should at least read F-actin by the way). I think the legend to Supp Fig.2 should be more precise to not induce us into erroneous interpretation of Fig.3, which has obviously not changed (?).

The original images that were used to illustrate the technique in Fig. 3 were not representative of the morphology under LCX treatment. We have now replaced the Fig. 3 images with ones that are more representative. In addition, the supplemental figure is now Supp. Fig. 3 and we have corrected the figure legend. We think these should adequately illustrate the phenomenon. 

- The discussion has been indeed improved. It does however not deal with the question

where the effect on the actin cytoskeleton of LCX vs. HCX treatment may attack. Could

there be an effect already during GAG synthesis in Golgi/cytosol and subsequent PG

transport, or is there only an effect on (extracellular) signaling via secreted or

membrane-bound PGs? (Let's say that is a question that would interest me personally,

but you need not answer it).

This is also a great question. I actually lean towards the idea that the xylosides might be acting on the cytoskeleton totally independentely of GAGs, since the effects of LCX are modest on GAG secretion (though they do change GAG composition). However, proving this is beyond the scope of the current manuscript, and I did not want to put idle speculation in the text. 

- There are still some errors and typos, see below.

Line numbers refer to the Word (.docx) document "Final Revision" !

Abstract: the reformulated lines 29-31 (-32) are not very "elegant", and the end may be

misleading ("higher concentrations had minor effects"). I'd propose something like:

To our surprise, we found that concentrations of xylosides in the nanomolar to micromolar

range had major effects on cell morphology of hippocampal neurons as well as of Neuro2a

cells, affecting both actin and tubulin cytoskeletal dynamics. Such effects/morphological

changes were not observed with higher xyloside concentrations.

Thanks for taking so much time for this. We have reworded using your suggestions.

Line32: Xylosides ... produces...

Line33: ...large change in GAG chain synthesis rate

Finally, you did not include the effect on GAG composition in the Abstract, why? You

have done so in the Introduction, where you state that your study may contribute to

understanding "how a minor shift in GAG composition can affect biological processes..."

Cell culture: Stock solutions are now described, but it would not have been necessary to

do it two times (lines 93, 107).

We now include GAG composition I the abstract. 

Line108: 10 m should probably read 10 min.

Growth cone analysis:

Lines164-167: this is a bit unclear for the reader. "The randomized files were then

numbered sequentially and saved for reference. Duplicates of these files had all identifying

information removed and then the numbered files were analyzed."

I guess that the duplicate files without information were analyzed (to make for 'double

blind')? Here it sounds as if the reference files (the "numbered" ones) were analyzed. I'd

prefer a simple: "Analysis was then performed on duplicates of these files from which all

identifying information had been removed".

We again thank the reviewer for suggesting better wording.

Lines162- : this does not really answer my question how "collapsed" GC were counted?

(as seen in Fig.1: LCX shows a neat, large GC, DMSO an almost collapsed, and HCX no

visible GC at all, making it impossible to count/evaluate microtubules).

We divided neurons into categories based on whether they contained looped microtubules or not in any of their growth cones, and did not focus on any other morphological features, including collapse. Low power images show that most growth cones had some actin staining suggesting that they are not collapsed, but these images do not have the resolution for further evaluation. We hope that the current presentation focusing on microtubules in growth cones is sufficient to demonstrate the phenomenon.

Results:

Not very "elegant" beginning: "we sought to..., but we sought to..."; and the first sentence

is not really true since you did not want to inhibit GAG synthesis here. Maybe you could

start with something like "Previous studies on GAG chain synthesis had used...

Here, we wanted to establish a dose-response curve... to determine...".

I’ve tried to rewrite this sentence, but it seems that being accurate is not entirely compatible with being elegant. So I changed it to “xyloside treatment”, rather than GAG chain inhibition.

Fig.1: You could have at least added F-actin staining of those "real" neurons since the

rest of the paper is mostly about lamellae and F-actin on N2A cells. F-actin is shown in

Supp Fig.1, but there an HCX image is missing.

Unfortunately, we concentrated on MT looping when we took these images, and so we did not have suitable images that contain F-actin for HCX at the equivalent power. I am including a reviewer’s figure that does have f-actin staining of hippocampal neurons, but these images were taken at 10X, such that microtubule looping is poorly imaged. But it does show f-actin p-domains in most neurites. We would be happy to include this figure if the reviewer thought it will help. 

Fig.2: As in the text you say that velocity and total distance were significantly different,

this should be shown in the figure (that shows only velocity), or at least the significance

values for total distance mentioned in the text.

We have now included the total distance in the results, though the result is not really independent since the persistence was not significantly different.

Fig.3: the image selected in Supp Fig.2 is a bit closer in comparison now, but I'm still not

convinced: what exactly do you designate lamellipodium here in Fig.3 (clearly

identifiable in the Supp. Fig.2 for the LCX cell). 

We should see (if I get it right??) that in LCX treated cells there are well-formed lamellipodia (reminiscent of neural growth cone), but less and thinner actin bundles than in HCX cells.

We agree that the images were not optimal As noted above, we have replaced the images in Fig. 3 with ones more representative. They closely align with what we find in the supplementary figure (Now Supp. Fig. 3).

Several typos in Supp Fig.2 legend on the figure itself (but not in the manuscript).

Line 273: (Figure 2C) should read Fig. 3C.

Discussion:

Lines 303-4: "Treatment with xyloside treatment..."

Line 408: I don't see how you can suggest a different action of LCX on hippocampal

neurons and N2A cells based on MT looping in growth cones, since the latter don't form

a growth cone (at least not in your study).

Thanks for pointing this out. I have reworded the paragraph.

Line 442: ...caused changed...

Line 443: predominant effect of actin, or on actin?

Line 452: full stop missing.

Line 454: "...the phenomenology is dose dependent". Normally, phenomenology is a

science (sort of) and cannot be dose dependent.

---

## [Editor Report · Decision Letter 2]

2 Jun 2022

A Novel Cytoskeletal Action of Xylosides

PONE-D-22-03467R2

Dear Dr. Geller,

We’re pleased to inform you that your manuscript has been judged scientifically suitable for publication and will be formally accepted for publication once it meets all outstanding technical requirements.

Kind regards,

Catherine FAIVRE-SARRAILH

Academic Editor

PLOS ONE
---

## [Editor Report · Acceptance letter]

20 Jun 2022

PONE-D-22-03467R2 

A Novel Cytoskeletal Action of Xylosides 

Dear Dr. Geller:

I'm pleased to inform you that your manuscript has been deemed suitable for publication in PLOS ONE. Congratulations! Your manuscript is now with our production department. 

Kind regards, 

on behalf of

Dr. Catherine FAIVRE-SARRAILH 

Academic Editor

PLOS ONE